# Synergism Antiproliferative Effects of Apigenin and Naringenin in NSCLC Cells

**DOI:** 10.3390/molecules28134947

**Published:** 2023-06-23

**Authors:** Xiongxiong Liu, Ting Zhao, Zheng Shi, Cuilan Hu, Qiang Li, Chao Sun

**Affiliations:** 1Institute of Modern Physics, Chinese Academy of Sciences, Lanzhou 730000, China; lxx002@impcas.ac.cn (X.L.);; 2Key Laboratory of Heavy Ion Radiation Biology and Medicine, Chinese Academy of Sciences, Lanzhou 730000, China; 3Key Laboratory of Basic Research on Heavy Ion Radiation Application in Medicine, Gansu Province, Lanzhou 730000, China; 4University of Chinese Academy of Sciences, Beijing 100049, China

**Keywords:** apigenin, naringenin, non-small cell lung cancer, synergism, antiproliferation

## Abstract

Non-small cell lung cancer (NSCLC) is one of the leading cancer killers. Apigenin (Api) and Naringenin (Nar) are natural bioactive substances obtained in various vegetables and fruits, possessing anti-tumor effects across multiple studies. This study investigated the latent synergistic antiproliferative functions of Api and Nar in A549 and H1299 NSCLC cells. Cell viability was determined after incubating with different concentrations of Api, Nar, or the combination of Api and Nar (CoAN) for 24 h. Analysis using the CompuSyn software revealed that the CI value of each combined dose was < 1, depicting that the two drugs had a synergistic inhibitory effect. The CoAN (A:N = 3:2) group with the lowest CI value was selected for subsequent experiments. The IC_50_ of CoAN (A:N = 3:2) was used to determine the cell cycle, the expression ratio of Bax to Bcl2, Caspase 3 activity, and mitochondrial function to assess oxidative stress and apoptosis. The results established that CoAN treatment caused significant cytotoxicity with cell cycle arrest at G2/M phases. Furthermore, CoAN significantly enhanced mitochondria dysfunction, elevated oxidative stress, and activated the apoptotic pathway versus Api or Nar alone groups. Thus, the CoAN chemotherapy approach is promising and deserves further research.

## 1. Introduction

Non-small cell lung cancer (NSCLC) is a highly aggressive and lethal tumor. Despite significant advancements in therapeutic strategies, NSCLC is responsible for endangering various lung cancer patients. GLOBOCAN 2020 data revealed that the incidence of lung cancer (11.4%) is second to breast cancer (11.7%), which is the leading cause of cancer mortality worldwide (1.8 million deaths) [1]. Treatment of NSCLC requires a combination of multiple therapeutic methods, such as chemotherapy, targeted therapy, and immunotherapy. However, optimal treatment for NSCLC depends on the staging, molecular subtype, and physical condition of the patients [2,3,4,5]. Almost all patients with NSCLC require chemotherapy. Although there have been advancements in NSCLC chemotherapeutics, the low selective toxicity blocks their clinical application. Therefore, we need novel drugs and approaches targeting NSCLC with minimal side effects.

Flavonoids are biologically active components primarily extracted from plants [6]. They possess the roles of anti-inflammatory, antioxidant, hepatorenal-protective, neuro-protective, anticancer, etc. [7,8]. Moreover, Mazu-rakova et al. [9] reported that flavonoids could potentially treat hypertensive disorders during pregnancy, which is necessary for the health of the mother and child. Apigenin (Api) and Naringenin (Nar) are significant flavonoids with anti-tumor effects that coexist in various fruits and vegetables, including berries, chamomile, citrus fruit, onions, and broccoli [10,11]. Api possesses beneficial and health-promoting properties due to its low intrinsic toxicity and remarkable anticancer effects, particularly in hepatocellular carcinoma, prostate cancer, and lung cancer [10,12]. Moreover, Api is a multi-targeting agent that regulates critical signaling pathways such as Akt/mTOR, JAK/STAT, NF-κB, etc. These pathways are involved in cancer development and progression [13]. Similarly, Nar can inhibit cancer progression via multiple mechanisms, including apoptosis induction, cell cycle arrest, angiogenesis hindrance, and by modifying various signaling pathways, including Wnt/β-catenin, PI3K/Akt, NF-ĸB, etc. [11]. However, high concentrations of flavonoids are needed to produce ideal tumor-killing effects [11,14]. Researchers have assumed that the combinations of lower concentrations of Nar and Api may provide synergistic effects. This could heighten cytotoxicity in malignant cells, and several plant constituent flavonoids can synergistically inhibit tumor cell proliferation.

Hence, this study was designed to determine whether Api and Nar had synergistic effects on the NSCLC A549 and H1299 cell lines. We detected cell proliferation, cell cycle, oxidative stress, mitochondrial function, and NSCLC cell apoptosis as the biological endpoint.

## 2. Results

### 2.1. Cell Viability

The antiproliferative effect of Nar, Api, or CoNA was determined on A549 and H1299 cells using the CCK8 assay. A549 and H1299 cells were incubated for 24 h with 0–100 μM of Api or 0–400 μM Nar, respectively. As represented in Figure 1A, 6.25, 12.5, 25, and 50 μM of Nar showed the effects of promoting H1299 cell growth. However, the stimulatory effects were reduced with increasing Nar concentration and returned to basal level at 100 μM of Nar. The IC50 values of Nar were 143.3 ± 3.1 μM and 370.1 ± 4.9 μM for A549 and H1299 cells, respectively. The IC50 values of Api were 45.7 ± 1.3 μM and 69.8 ± 2.5 μM in the A549 and H1299 cells, respectively. However, CoAN treatment elevated toxicity in A549 and H1299 cells more than either the Nar or Api alone. At a 3:2 (A:N) combination ratio, the IC50 was 28.7 ± 0.9 μM and 32.5 ± 3.9 μM for A549 and H1299 cells, respectively. These were significantly lower than either 1:1 or 2:3 ratios (Figure 1C,D). 

Moreover, we observed the A549 and H1299 cells after 28.7 μM and 32.5 μM of Api, Nar, or both (CoAN, A:N = 3:2) treatments for 24 h, respectively, followed by observing and imaging the growth state of the cells under the inverted microscope. As shown in Figure 1E, the cell morphology was normal, but the number decreased in the Api and Nar groups compared to the control. However, the CoAN groups had reduced cell numbers, enlarged volumes, and the occurrence of vacuoles compared to the control. 

### 2.2. Synergistic Effect of Api and Nar

Isobolographic analysis helped determine the combined effects of Api and Nar. We prepared the solutions of Api and Nar combinations in ratios of 3:2, 1:1, and 2:3 to investigate possible synergism. The combination index (CI) values at IC_50_ of different combination ratios were determined with the CompuSyn software [15,16]. The CI value indicates the interaction degree between drugs. A CI near 1 of the combined compounds depicts an additive effect, <1 synergism, and >1 antagonism. Based on the results (Figure 2), the CI <1 in the Api and Nar ratios showed evident synergism. Moreover, the ratio of 3:2 (A/N) demonstrated more significant (*p* < 0.01) synergistic effects than either 1:1 (A/N) or 2:3 (A/N) ratios. Therefore, the 3:2 ratio became the optimal scheme for subsequent studies.

### 2.3. Cell Cycle Distribution

The cytotoxicity effect of chemotherapy drugs on cells is associated with cell cycle distribution. Therefore, the cell cycle induced by Api, Nar, or CoNA was determined. A549 and H1299 cells were incubated with 28.7 μM and 32.5 μM of Api, Nar, or both (CoAN) for 24 h, respectively, followed by PI staining. Flow cytometry helped quantify the cell populations in each phase of the DNA content. The results of the cell cycle phases are depicted in Figure 3. Table 1 shows the average statistical results of cell cycle phase percentages across three independent experiments of A549 and H1299 cell lines. Compared to the control, Api, or Nar alone groups, CoAN treatment significantly (*p* < 0.01) decreased the cell number in the G0/G1 phase of both cell lines. Nar and CoAN treatment significantly reduced the cell population in the S phase. Accordingly, the cellular percentage in the G2/M phase accumulated in A549 and H1299 cells (*p* < 0.01) corresponding to their respective control, Api, or Nar alone groups.

### 2.4. CoAN Aggravated the Cell Apoptosis

Cell apoptosis and Caspase 3 activity were further detected. The A549 and H1299 cells were treated using 28.7 μM and 32.5 μM of Api, Nar, or both (CoAN), respectively. The results in Figure 4A,B indicate that Api or Nar treatment elevated the cell apoptosis of both cells, compared to the control (*p* < 0.01). Moreover, CoAN treatment significantly upregulated the apoptotic rate (Table 2). The treatment also enhanced Caspase 3 activity in both cell lines (*p* < 0.01) (Figure 4C,F), corresponding to either Nar or Api alone. Furthermore, Western blot assays helped detect the Bax, Bcl-2, and cleaved Caspase 3 proteins in A549 and H1299 cells to determine the apoptotic pathway. As shown in Figure 4D–H, CoAN treatment significantly elevated the Bax (pro-apoptotic protein) but reduced Bcl-2 (anti-apoptotic protein) proteins, significantly enhancing the ratio of Bax to Bcl-2. The cleaved-Caspase 3 expressions were significantly elevated compared to Api or Nar alone, consistent with previous results. Thus, the mitochondrial-related apoptotic pathway was activated.

### 2.5. CoAN Enhanced Lipid Peroxidation and ROS Levels

The malonaldehyde (MDA) content was used to quantify lipid peroxidation. The MDA contents and ROS levels in A549 and H1299 cells were determined after 24 h of treatment with 28.7 µM and 32.5 µM of Api, Nar, or both (CoAN), respectively. As shown in Figure 5A,B, Api or Nar treatment did not significantly (*p* > 0.05) improve the ROS levels in the cell lines compared to the control. The MDA concentration was not (*p* > 0.05) elevated in A549 cells. However, it was significantly (*p* < 0.05) enhanced in H1299 cells using Nar treatment, compared to the controls. Moreover, compared to the controls, MDA concentrations were remarkably improved using Api treatment in both cell lines. Furthermore, the CoAN treatment significantly (*p* < 0.01) increased ROS levels and MDA contents in the cell lines relative to the control, Api, or Nar.

### 2.6. CoAN Promotes Mitochondrial Dysfunction in NSCLC Cells

Mitochondria are the prominent energy-producing organelle of eukaryotic cells. The mitochondrial membrane potential (MMP) change and ATP production can assess the mitochondrial function within cells. A549 and H1299 cells were treated with 28.7 µM and 32.5 µM of Api, Nar, or both (CoAN) for 24 h, respectively. As shown in Figure 6A,B, Api or Nar treatment significantly (*p* < 0.01) decreased the MMP in both cell lines compared to the control. Intriguingly, CoAN treatment significantly (*p* < 0.01) decreased MMP compared to Api or Nar alone (Figure 6C). ATP levels were further analyzed, indicating that CoAN treatment (*p* < 0.01) reduced the ATP synthesis compared to the controls in both cell lines, Api or Nar alone (Figure 6D). Therefore, CoAN promotes mitochondrial dysfunction leading to cell apoptosis.

## 3. Discussion

Different studies have demonstrated that Api or Nar possess potential antitumor functions with varying molecular mechanisms [11,17]. However, most studies have incorporated high Api or Nar concentrations to obtain an effective inhibitory effect. Thus, the median Api and Nar concentrations were 50 µM and 100 µM, respectively [11,14]. Both Nar and Api are insoluble in water. DMSO could dissolve Nar and Api to ameliorate their cellular consumption. Ideally, combining different agents with similar bioactivity improves the therapeutic efficacy and reduces the toxicity of individual agents. Intriguingly, CoAN at a 3:2 ratio had evident synergistic effects on both A549 and H1299 cells. The IC50 value was the lowest concentration compared to 2:3 and 1:1 ratios, and Api or Nar alone. The isobologram analysis demonstrates that Api and Nar have synergistic antiproliferative effects on A549 and H1299 cells. NSCLC cells A549 (p53+/+) and H1299 (p53−/−) have different phenotypic characteristics. According to the above data, CoAN has synergistic antiproliferative efficacies on p53 wild type NSCLC (A549) and p53-deficient NSCLC (H1299) cells. We speculated that the antiproliferative effects exert a p53-independent pathway. 

Cell cycle arrest is one of the indicators by which to assess the efficacy of chemotherapy drugs to treat cancers. Therefore, we investigated the cell cycle arrest capacity of Api, Nar, or CoAN. Choi et al. [18] reported that 50 or 100 μM Api caused the arrest of the cell cycle of breast cancer at the G2/M phase. These findings were consistent with our results that Api significantly (*p* < 0.05) blocked A549 and H1299 cells in the G2/M phase. Ahamad et al. [19] reported that 100 µM of Nar significantly arrests the hypodiploid cells in the G0/G1 phase. However, both A549 and H1299 cells were treated with 28.7 µM and 32.5 µM of Nar alone, having no significant (*p* > 0.01) effects on the number of cells in the G0/G1 phase. Furthermore, CoAN treatment (*p* < 0.01) considerably induced the number of cells accumulated at the G2/M phase in the cell lines, along with cell population reductions in the G0/G1 and S phases.

Additionally, the apoptotic rates of CoAN on A549 and H1299 cells were determined. Previous studies showed that Nar and Api could induce cell apoptosis [20,21]. Consistent with our results, both Nar and Api treatments significantly (*p* < 0.01, compared to the respective control) elevated the apoptosis (early stage + late stage apoptosis) of both cell lines (Table 2). Furthermore, CoAN treatment significantly increased the total apoptosis rates (*p* < 0.01) relative to Api or Nar alone. Bcl-2 protein family members are considered vital apoptosis monitors, and Bcl-2 is one of the essential anti-apoptotic proteins [22]. It could prevent apoptosis by binding and repressing Bax activity [23] or inhibit the caspase functions [24]. In contrast, an activated Bax can form holes in the outer membrane of mitochondria. Once cytochrome C is released from the Bax channels, apoptosis occurs [25]. The ratio of Bax to Bcl-2 has positively regulated apoptosis. Excess Bax can potentially cause apoptosis [26]. Therefore, Bcl-2 is a prognostic marker for lung cancer [27]. Moreover, Alam M et al. [28] indicated that the EGFR signal could regulate the interactions between Bax and Bcl-2 during NSCLC apoptosis. Our results revealed that the CoAN treatment enhanced the values of Bax/Bcl-2 (*p* < 0.01) compared to either Api or Nar alone in both cell lines (Figure 4E,H). In addition, Caspase-3 is a primary regulator in cell apoptosis, the ultimate enforcer in apoptotic death. Caspase-3 usually exists as a zymogen (32 kDa) and is activated to p17 or p11 subunits using initiator caspases during apoptotic flux [29]. Our results demonstrated that both Api and Nar alone could significantly (*p* < 0.05) improve the activity of Caspase-3. However, CoAN treatment significantly (*p* < 0.01) increased the level of cleaved Caspase-3 (p17 subunit) compared to Api or Nar alone. Therefore, the increased Caspase-3 levels ultimately enhanced NSCLC cell apoptosis.

The overproduction of intracellular ROS is one of the leading drivers causing apoptosis [30,31]. ROS could destroy the mitochondrial membrane and form pores, inducing the Cytochrome-c permeability to penetrate the cytoplasm. This prompts the caspase function, for example, Caspase-3, that results in cell apoptosis [32,33,34]. Moreover, flavonoids have pro-oxidant effects in tumor cells but display antioxidant effects in normal cells. This could contribute to tumor cells with higher substrate oxidation levels than normal cells [35]. The structure of Api does not have catechol hydroxyl groups, which determines its antioxidative ability [36]. Several studies have reported that Api enhanced intracellular ROS generation and produced pro-oxidative effects across various tumor cells [37,38,39,40]. Nar has the characteristics of most flavonoids that could effectively and selectively enhance oxidative stress and apoptosis in tumor cells [41]. In our study, neither Api nor Nar caused evident (*p* < 0.05) increases in ROS production in the cell lines. Intriguingly, the same concentration of CoAN treatment elevated the ROS and MDA (*p* < 0.01) compared to the control, Api, or Nar alone. Thus, the results indicate a synergistic effect leading to oxidative stress.

Mitochondria are essential energetic cellular organelles [42]. Excessive ROS production by the cells could lead to mitochondrial membrane damage and induce apoptosis. In contrast, dysfunctional mitochondria could also increase oxidative stress [43,44,45]. Our results indicated that NSCLC cells treated with Api and Nar alone exhibited a significant (*p* < 0.01 compared to the control) reduction in MMP. The results were evidenced by the value of red/green fluorescence, depicting depolarized mitochondria. Moreover, CoAN treatment induced a significant (*p* < 0.01) MMP reduction in both NSCLC cell lines, compared to their respective Api or Nar alone group. Simultaneously, the study observed that the contents of ATP decreased (*p* < 0.05) in the Api and Nar alone treatment groups. Intriguingly, the equivalent of CoAN treatment evidently (*p* < 0.01) down-regulated the ATP levels in both cell lines, compared to their respective Api or Nar alone groups. The above findings established that the CoAN had synergistic effects on NSCLC cell viability, regulated through excessive ROS production. This induced a vicious cell cycle causing apoptosis. 

Furthermore, CoAN is a crucial factor so that an agent can selectively kill tumor cells but with a protective effect on normal tissues. Interestingly, flavonoids have differential effects on normal and tumor tissues [46,47,48,49,50,51,52,53,54,55,56,57,58,59,60,61]. For instance, Api has been associated with a reduced risk of diabetes. Api and Nar have been linked to improving cardiovascular health as well as anti-inflammatory and hepatorenal-protective effects. Moreover, they could be used to prevent and treat neurodegenerative diseases, including Alzheimer’s. Therefore, it was concluded that Api and Nar possess low toxicity and could selectively kill tumor cells. These properties have potential clinical value and are worth additional research to determine the synergistic antiproliferative effects of CoAN. 

Accordingly, CoAN has pronounced synergistic antiproliferative effects in NSCLC cells. These effects are compared to either Api or Nar alone groups by blocking the cells in the G2/M phase, increasing oxidative stress, promoting mitochondrial dysfunction, and finally, leading to cell apoptosis. Api and Nar can be protective of vertebrate models. Hence, more in vivo experiments are needed to verify the antitumor effects of CoAN. 

Although phenolic compounds are well known for their health benefits, they also possess dual effects on tumor cells, depending on the stage of tumor development. Cvorovic et al. [62] observed a dual role for anthocyanidins. They have antioxidant effects in tumor cells with low proliferation rates and low malignant potency. Antioxidants are used to protect cells from damage caused by ROS. However, clinical trials have provided inconsistent results. Sayin et al. [63] demonstrated that antioxidants reduced oxidative stress and DNA damage. However, they induced the mice to develop more tumors at more advanced stages and reduced survival in lung cancer. Zou et al. [64] reported that antioxidants increased tumor size in early neoplasias and tumor grades in more advanced lesions without impacting intestinal tumor initiation. Furthermore, administering some phenolic compounds can induce tumor growth since they can activate oncogenes [65,66]. These results indicated that applying phenolic compounds in cancer therapy should accurately distinguish pathological processes in the clinical trial stage, with an optimal treatment plan for patients. 

## 4. Materials and Methods

### 4.1. Cell Culture

A549 and H1299 cell lines were procured from the Cell Bank of the Chinese Academy of Sciences (Shanghai, China). The cells were incubated in RPMI medium 1640 (Sigma-Aldrich, Burlington, MA, USA), which was supplemented with 1% (*v*/*v*) penicillin-streptomycin and 10% (*v*/*v*) FBS (fetal bovine serum, Minhai, China). Under saturated humidity, the cells were maintained in a 5% CO_2_ constant temperature incubator (Thermo Electro Corporation, Waltham, MA, USA). 

### 4.2. Cell Treatments

We purchased Apigenin (pCode: S2262070005001; CAS number: S2262; lot ID: S226207) and Naringenin (pCode: S2394020002501; CAS number: S2394; lot ID: S239402) from Selleck Chemicals LLC. Dimethyl sulfoxide (DMSO, Sigma-Aldrich) (pCode: 102125675; CAS number: 67-68-5; lot ID: WXBC7821V) was utilized to prepare the mother solution of 100 mM and 50 mM of Api and Nar, respectively. Then, the cells were incubated with varying concentrations of Api, Nar, or CoAN (combining Api and Nar Api and Nar). CoAN concentration was determined as the sum of Api and Nar concentrations. For example, to prepare a CoAN solution with a concentration of 50 μM and an A/N ratio of 3:2, the concentrations of Api and Nar would be 30 μM and 20 μM, respectively. The DMSO (*v*/*v*) was less than 0.01%.

### 4.3. Cytotoxicity Assays

Cell viability was determined using the Cell Counting Kit-8 (CCK-8) (Solarbio, Beijing, China) assay. The cells were seeded in 96-well plates (1 × 10^4^ cells/well in 200 μL culture medium) and incubated for 24 h. Then, we changed the medium into 0–400 μM concentrations of Api or Nar or 0–70 μM (the sum of the two) of CoAN (combining Api and Nar). The combination ratios of Api to Nar were designed to be 3:2, 1:1, or 2:3, enabling the selection of the most optimal concentration. After a 24 h residence time, the CCK8 working solutions were added. The cells were continuously incubated for 1–4 h at 37 °C. Later, the light absorption value was determined at 450 nm in each group using a microplate reader (Epoch, Fremont, CA, USA). The ratio of absorption values helped represent cell survival, with the calculation formula of: (dosing − blank)/(control − blank).

### 4.4. Synergistic Effect of Api and Nar

The synergistic effect of Api and Nar was demonstrated using isobologram analysis. The CompuSyn software provided a CI value to illustrate the drug interactions [67]. The CI value was determined according to the Chou et al. [15,16]. CI near 1 indicated an additive effect, <1 synergism, and >1 antagonism of the drug combination.

### 4.5. Cell Cycle Analysis

The collected cells were centrifuged, organized, and fixed using 70% cold ethyl al-cohol for over 48 h at −20 °C. Afterward, the cells were centrifuged (500× *g*, 4 min, at 4 °C) and washed using ice-cold PBS. The cell sediments were incubated for 30 min with 50 μL of RNase A (Sigma, 100 μg/mL) and kept on ice. Immediately, 100 μL of propidium iodide (PI, Sigma, 50 μg/mL) was added and stained for 30 min. Flow cytometry (Susmex CyFlow Cube 6, Japan) detected the cell population distribution. The data were analyzed with the FlowJo 7.6.1 software.

### 4.6. Detection of Mitochondrial Membrane Potential (MMP)

MMP was determined using the MMP detection kit JC-10 (Solarbio, China). The cells were planted in a φ35 culture dish and exposed to various treatments. Afterward, the cells were centrifuged and resuspended in a 500 μL culture medium. Then, 500 μL of JC-10 dyeing working solution was added, and incubated for 30 min in a cell incubator. Then, the cells were harvested and washed with a staining buffer. Flow cytometry helped measure the fluorescence intensity. The MMP was determined by the red-to-green fluorescence ratio and analyzed using Flowjo_V10.

### 4.7. Reactive Oxygen Species (ROS) Detection

The treated cells were harvested and washed using RPMI 1640 (serum-free, phenol red-free). Then, 2,7-dichlorofluorescin-diacetate (5 μM, DCFH-DA; Sigma-Aldrich) were put in and reacted without light at 37 °C for 30 min. Afterward, the cells were centrifuged and collected. The pellets were resuspended in PBS. Finally, the fluorescence values of ROS were detected with flow cytometry. 

### 4.8. Analysis of ATP Levels

According to the manufacturer’s instructions, the ATP levels were measured with the ATP Detection Kit (Solarbio, China).

### 4.9. Detection of Apoptosis

Caspase 3 activity indicates the cell apoptosis induction, measured using the Caspase 3 Activity Assay Kit (Bioworld, Nanjing, China). The results were determined as a ratio relative to the control.

Following the recommendation, cell apoptosis was detected using an Annexin V-FITC/PI apoptosis detection kit (BD, USA). The cells were obtained using centrifugation and washed with ice-cold PBS, followed by resuspension in 400 μL of binding buffer. The cells were incubated at room temperature in the dark for 15 min after adding Annexin V-FITC (5 μL) and PI solution (10 μL) and analyzed using flow cytometry.

### 4.10. Western Blot Analysis

Western blot assay was conducted according to the reported literature [68]. Then, the treated cells were lysed with RIPA buffer, and the protein was collected. The 10% SDS-PAGE gel was selected, which helped finish the experimental procedures such as electrophoresis and transmembrane. The primary antibodies Bax (1:3000, Proteintech, Wuhan, China), Bcl-2 (1:3000, Proteintech), GAPDH (1:10,000, Proteintech), and cleaved Caspase 3 (1:1000, Cell Signaling Technology, Danvers, MA, USA) were incubated overnight at 4 °C. Finally, the second antibody was labeled with horseradish peroxidase and incubated for 1 h. The membrane was washed thrice using PBST and exposed to a Protein Imaging System (Amersham Imager 680, Logan, UT, USA). 

### 4.11. Statistical Analysis

The data were obtained from three independent experiments, representing ± SEM. All of the statistical analyses were performed with SPSS Statistics Version 20.0 software (SPSS Inc., Chicago, IL, USA). The Student’s t-test was used for comparing the mean values between the two groups, and one-way ANOVA was utilized to compare multiple groups. *p*-values < 0.05 were considered statistically significant.

## 5. Conclusions

The CoAN of Api and Nar have hopeful synergistic antiproliferative functions in NSCLC cells. The data support that a plant-enriched diet is associated with a reduced incidence rate of tumors and the prevention of various cancers. We hope that combined Api and Nar could become anti-tumor agents to antagonize NSCLC cancer. Additional in vivo and randomized clinical trials can validate the synergistic functions of Api and Nar in various cancer types, including NSCLC.

## Figures and Tables

**Figure 1 molecules-28-04947-f001:**
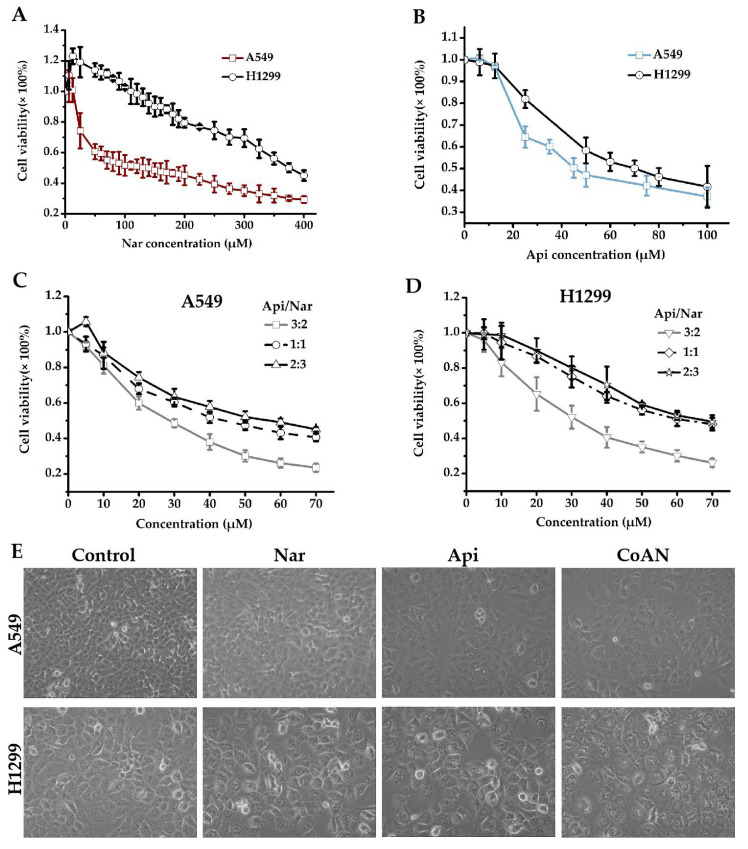
The cell viability detection. (**A**–**D**) involve the cell cytotoxicities measured with CCK8 assay. A549 (**A**) and H1299 (**B**) cells were incubated for 24 h using 0–100 μM of Api or 0–400 μM Nar, respectively. A549 (**C**) and H1299 (**D**) cells were incubated for 24 h using 0–70 μM of CoAN (the Api to Nar ratio was 3:2, 1:1, or 2:3), respectively. (**E**) The morphology of A549 and H1299 cells after 28.7 μM and 32.5 μM of Api, Nar, or both (CoAN, A:N = 3:2) treatments for 24 h, respectively. Api: Apigenin, Nar: Naringenin.

**Figure 2 molecules-28-04947-f002:**
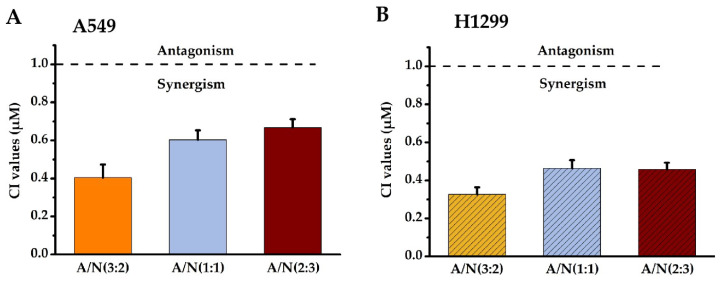
The synergistic effects of Api and Nar were demonstrated with isobologram analysis. CI values of Api and Nar across different combined treatment groups in A549 (**A**) and H1299 cells (**B**). A CI near 1 depicts additive, <1 synergism, and >1 antagonism effects. A: Apigenin, N: Narigenin, CI: combination index.

**Figure 3 molecules-28-04947-f003:**
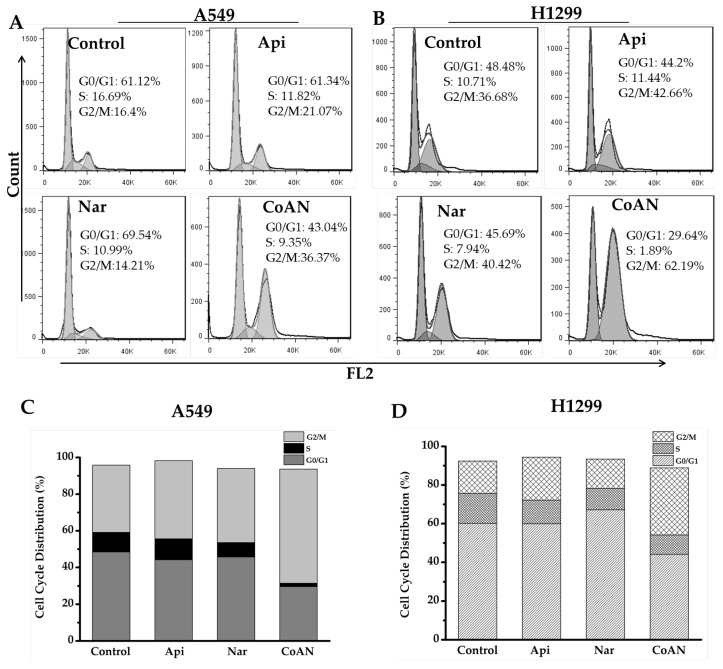
The cell cycle analysis of the A549 and H1299 cells after 28.7 μM and 32.5 μM of Api, Nar, or both (CoAN) treatments for 24 h, respectively. (**A**,**B**) are representative diagrams of the cell cycle phases. (**C**,**D**) are the statistical data of the cell cycle phase distributions in A549 and H1299 cells.

**Figure 4 molecules-28-04947-f004:**
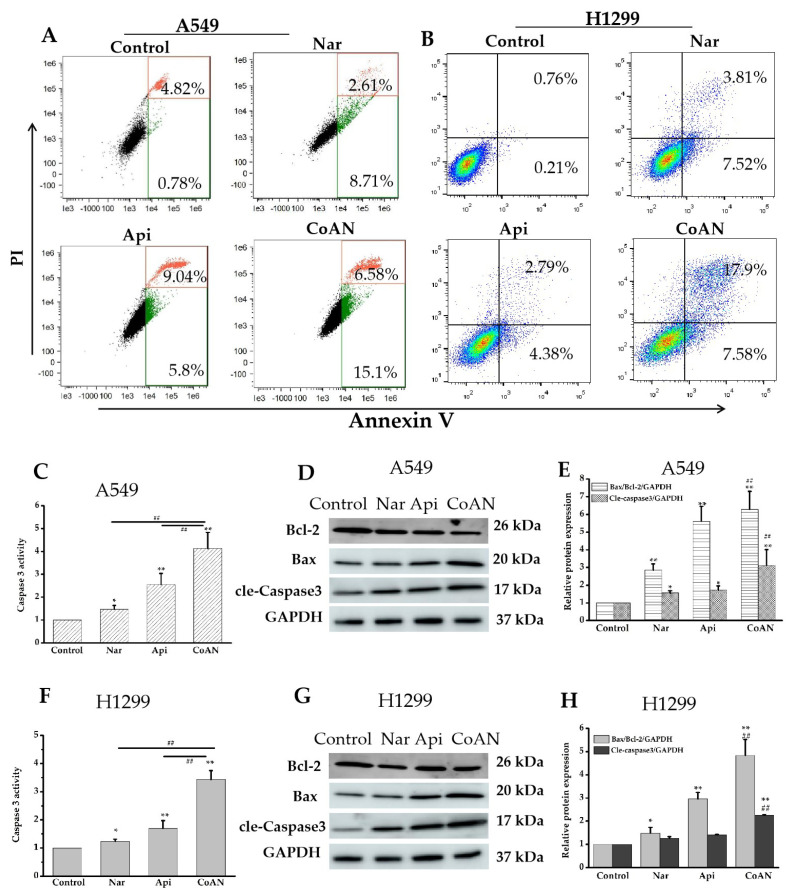
The cell apoptosis detection. A549 and H1299 cells were exposed to 28.7 µM and 32.5 µM of Api, Nar, or both (CoAN), respectively. (**A**,**B**) are the graphs representing apoptosis detected using flow cytometry. (**C**,**F**) are the relative Caspase 3 activities. (**D**,**G**) are the Bcl-2, Bax, and GAPDH protein expression products. (**E**,**H**) are the statistical data of the gray value of Bax/Bcl-2/GAPDH and cleaved-Caspase 3/GAPDH. * *p* < 0.05 and ** *p* < 0.01 vs. control. ## *p* < 0.01 vs. Api or Nar.

**Figure 5 molecules-28-04947-f005:**
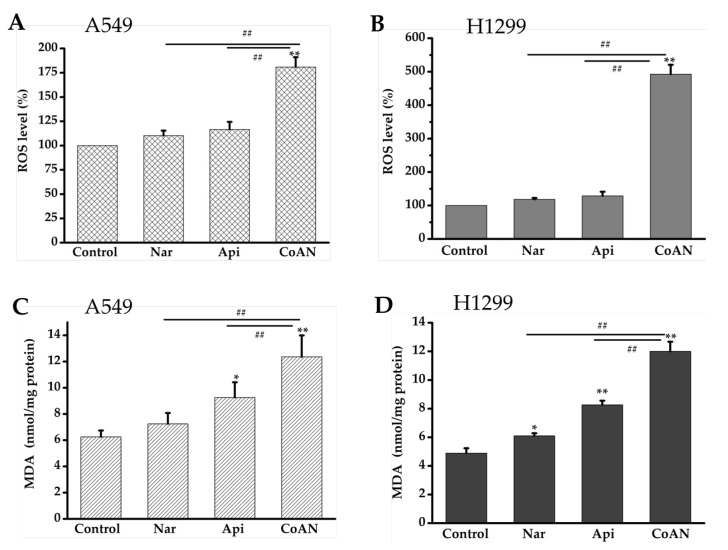
CoAN elevated MDA and ROS levels in NSCLC. A549 and H1299 cells were exposed to 28.7 µM and 32.5 µM of Api, Nar, or both (CoAN), respectively. (**A**,**B**) ROS levels; (**C**,**D**) MDA concentrations. * *p* < 0.05 and ** *p* < 0.01 vs. control. ^##^
*p* < 0.01 vs. Api or Nar.

**Figure 6 molecules-28-04947-f006:**
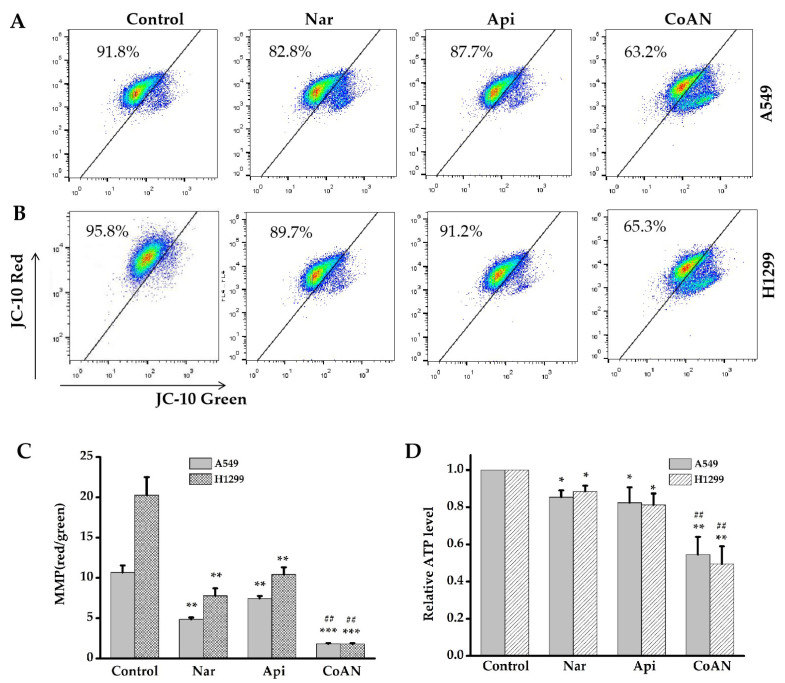
The MMP and ATP levels were determined after exposing A549 and H1299 cells to 28.7 µM and 32.5 µM of Api, Nar, or CoAN for 24 h, respectively. (**A**,**B**) represent the graphs of MMP detected with flow cytometry. (**C**) Statistical analysis of red/green fluorescence ratio. (**D**) The relative MMP levels. * *p* < 0.05, ** *p* < 0.01 and *** *p* < 0.001 vs. control, ^##^ *p* < 0.01 vs. Api or Nar. MMP: Mitochondrial membrane potential.

**Table 1 molecules-28-04947-t001:** The distribution of cell cycle phases (%).

	A549			H1299		
	G_0_/G1	S	G_2_/M	G_0_/G1	S	G_2_/M
Control	60.2 ± 0.88	15.5 ± 1.09	16.7 ± 0.44	47.99 ± 0.71	10.83 ± 0.61	36.34 ± 0.78
Api	59.96 ± 1.59	12.21 ± 0.35	22.21 ± 1.08 *	43.67 ± 0.68	12.21 ± 0.73	42.59 ± 1.34 *
Nar	67.21 ± 2.13	11.1 ± 0.14 *	15.12 ± 0.88	45.6 ± 0.66	8.18 ± 0.44 *	40.52 ± 0.69 *
CoAN	44.09 ± 1.08 **^##^	10.05 ± 0.63 *	34.78 ± 1.50 **^##^	28.3 ± 1.67 **^##^	2.12 ± 0.24 **^##^	61.38 ± 0.80 **^##^

* *p* < 0.05, ** *p* < 0.01 versus (vs.) Control; ^##^ *p* < 0.01 vs. Api or Nar.

**Table 2 molecules-28-04947-t002:** The apoptotic rate of cells (%).

	A549	H1299
Control	4.80 ± 0.72	1.10 ± 0.15
Api	14.21 ± 1.20 **	8.24 ± 0.95 **
Nar	9.82 ± 1.42 **	11.11 ± 0.19 **
CoAN	23.15 ± 1.38 **^##^	24.42 ± 0.96 **^##^

** *p* < 0.01 vs. Control; ^##^ *p* < 0.01 vs. Api or Nar.

## Data Availability

Not applicable.

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
