# Peer review of "Synergism Antiproliferative Effects of Apigenin and Naringenin in NSCLC Cells"

_molecules, 2023, doi:10.3390/molecules28134947_

Round 1
Reviewer 1 Report
There are reports that phenolic compounds can promote the development of tumors if they are administered at a certain stage of tumor development, consider it for introduction or discussion, even if you prove otherwise
Check grammar
Author Response
Please see the attachment, thank you!

Reviewer 2 Report
The article molecules-2450743 investigated the synergistic antitumor effects and the underlying apoptotic mechanisms of Apigenin (Api) and Naringenin (Nar) in A549 and H1299 non-small cell lung cancer (NSCLC) cells in vitro. The results demonstrated that the combined treatment with Api and Nar (3:2 at molar concentration) (CoAN) possessed the strongest synergistic inhibitory effect. And the CoAN treatment also resulted in the cell cycle arrest at G2/M phases and cell apoptosis via inducing the mitochondria dysfunction and oxidative stress. Flavonoids are rich in most of plant resources and have a wide range of biological effects. The flavonoid compounds in natural plant resources are all complex systems, and the interaction between different flavonoids is worth further investigation.This article provides an interesting perspective for understanding the synergistic antitumor activity of natural flavonoids.
Further revised the article according the following questions will help improve this article.
1. The theoretical basis for the investigation of the two flavonoids Api and Nar in the paper is not introduced. Do these two compounds coexist in the same plant resource with anticancer activity. And if so, what is their natural ratio?
2. Please define the concept of CoNA in detail when it first appears.
3. The reference [13] does not seem to cover the calculation method of The Combination Index (CI). Please proofread the reference and redescribe the calculation method of CI in this article.
4. In Figure 6 A and B, the horizontal and vertical coordinates should be annotated.
5. In Figure 6C, the basal membrane potential in control group of the two cell strains was significantly different. Is there any reasonable explanation on the mechanism behind the difference? And are there the similar result of difference in the ATP content between the two strains of cells in Figure 6D?
6. In LINE 190 and 191:“These findings were consistent with our results that Api significantly (p < 0.05) increased the numbers of both A549 and H1299 cells in the G2/M phase.”Here the description “increased the numbers of both ... cells”is not precise. In fact, it is the cells was blocked in this phase.
7. Further supplement of the photographs of typical morphological changes of cells during the treatment will help the reader to recognize of this research.
Author Response
Please see the attachment, thank you!
